# HIV/AIDS burden, attributable risk factors, and projections among reproductive-age adults in China, 1990–2035: A GBD 2023 analysis

Huibo Yan ⓘ *

Clinical Laboratory, Shanghai Putuo District Maternity and Infant Hospital, Shanghai, China

* yanhuibo0605@163.com

## Abstract

### Background

HIV/AIDS remains a leading infectious disease burden globally. While highly active antiretroviral therapy has transformed HIV into a manageable chronic condition, contemporary trends and attributable risk factors among reproductive-age adults in China—a critical demographic for epidemic control—remain poorly characterized. This study provides an updated comprehensive assessment of HIV/AIDS burden in this population using Global Burden of Disease Study 2023 data, which incorporates methodological refinements and post-pandemic data not available in earlier versions.

### Methods

We extracted age-standardised incidence, prevalence, mortality and disability-adjusted life-years (DALYs) for Chinese adults aged 15–49 years from GBD 2023. Joinpoint regression estimated annual percent change and identified significant trend inflection points. Population attributable fractions were analysed for unsafe sex, drug use, and intimate-partner violence. ARIMA models forecasted burden trends to 2035 with 95% uncertainty intervals.

### Results

Between 1990 and 2023, age-standardised rates increased substantially: incidence +301% (from 0.82 to 3.29 per 100,000), prevalence +774% (4.84 to 42.29), mortality +1,623% (0.13 to 2.24), and DALYs +1,439% (7.96 to 122.48). Males showed consistently steeper increases than females across all metrics (all p < 0.05). Unsafe sex accounted for 70% of HIV/AIDS mortality in 2023. ARIMA projections indicate that while male incidence will plateau at ~5.0 per 100,000 by 2035, female metrics will continue rising (incidence +21%, prevalence +30%), widening the sex disparity.

**Data availability statement:** All data underlying the findings described in this manuscript are fully available without restriction from the Global Burden of Disease Study 2023 (GBD 2023) public repository at https://vizhub.health-data.org/gbd-results/. The datasets used in this study were downloaded directly from this platform and are cited in the Methods section.

**Funding:** The author(s) received no specific funding for this work.

**Competing interests:** The authors have declared that no competing interests exist.

**Abbreviations:** AAPC, Average annual percent change; APC, Annual percent change; ARIMA, Autoregressive integrated moving average; ASDR, Age-standardised disability-adjusted life-year rate; ASIR, Age-standardised incidence rate; ASMR, Age-standardised mortality rate; ASPR, Age-standardised prevalence rate; BIC, Bayesian information criterion; DALYs, Disability-adjusted life years; GBD, Global Burden of Disease; HAART, Highly active antiretroviral therapy; MSM, Men who have sex with men; PAF, Population attributable fraction; PrEP, Pre-exposure prophylaxis; SDG, Sustainable Development Goals; UI, Uncertainty interval; UNAIDS, Joint United Nations Programme on HIV/AIDS.

## Conclusion

China's HIV epidemic has transitioned to a chronic disease burden. Notably, this burden shows pronounced sex differences. Mortality is rising among people living with HIV, particularly among women. These trends highlight urgent needs for gender-differentiated prevention strategies. Our findings provide measurable targets for China's National HIV/AIDS Action Plan (2024–2030) and SDG 3.3 monitoring. They further demonstrate how standardized burden estimation supports precision public health.

## 1. Introduction

Acquired immunodeficiency syndrome (AIDS) is caused by the human immunodeficiency virus (HIV). HIV/AIDS has caused an estimated 40 million deaths since 1980. It remains a leading global health threat [1]. The universal roll-out of highly active antiretroviral therapy (HAART) has fundamentally changed the disease trajectory. HIV infection is now a manageable chronic condition rather than a fatal disease. Life expectancy has increased markedly, and the epidemiological profile has shifted toward an ageing, chronically infected population [2]. Accurate quantification of this evolving burden is essential. Such data enable evaluation of programme performance and guide reallocation of prevention and care resources. In 2023, 39.9 million people were living with HIV worldwide and 630,000 died from AIDS-related illnesses [3]. In response, the United Nations has prioritized HIV in both the Millennium Development Goals (MDG 6) and the Sustainable Development Goals (SDG 3.3). Global investment reached US$ 500 billion between 2000 and 2015 to reduce incidence and expand treatment access. Despite these investments, China reported nearly 1.3 million people living with HIV by the end of 2023, where incidence has plateaued but prevalence and mortality continue to rise [4]. Whether the national strategy should continue to focus on incident infections or shift toward managing the growing chronic-care demand remains unclear. To inform this decision, we analysed 1990–2023 HIV/AIDS burden in China and projected trends to 2035 using the Global Burden of Disease 2023 study [5].

The Global Burden of Disease Study (GBD) provides a standardized, comparative framework for tracking HIV/AIDS across 204 countries and territories [5]. Earlier GBD analyses (2019, 2021) have documented national HIV trends in China [6–8], yet three critical gaps persist. First, these studies predate the COVID-19 pandemic and associated disruptions to HIV testing and treatment services, necessitating updated estimates through 2023. Second, they did not specifically focus on reproductive-age adults (15–49 years), a demographic that accounts for the majority of new diagnoses in China—males in this age group alone contributed 79.24% of reported cases in 2022 [9], and representing a strategic target prevention interventions including pre-exposure prophylaxis (PrEP) and harm reduction programs. Third, GBD 2023 incorporates methodological refinements—including updated HIV progression models and expanded data sources from 2020–2023—that yield more precise estimates for recent trends.



Compared with GBD 2021, GBD 2023 offers three key advancements relevant to HIV burden estimation: updated Cause of Death Ensemble modeling (CODEm) with expanded vital registration data from China's disease surveillance points system; revised risk factor attribution methods incorporating systematic reviews through 2023; and inclusion of post-COVID-19 pandemic data (2020–2023) that capture potential disruptions to HIV testing and treatment services. These improvements yield more precise estimates for the most recent period and enhance the reliability of projections to 2035.

Using GBD 2023, we examined national and sub-national trends in age-standardized incidence (ASIR), prevalence (ASPR), mortality (ASMR), and disability-adjusted life-years (ASDR) specifically for this key demographic, quantifying the contribution of age and sex. Joinpoint regression estimated annual percentage changes, while ARIMA models projected burden to 2035. Accordingly, this study provides an updated comprehensive assessment of HIV/AIDS burden, attributable risk factors, and projections specifically among reproductive-age adults in China, leveraging the most recent GBD 2023 methodologies to inform targeted prevention strategies and progress toward Sustainable Development Goal 3.3. These findings will inform the transition from an "incidence emergency" to a "chronic-care burden" framework, guiding precision prevention, early case-finding, and long-term chronic-disease management strategies tailored to China's reproductive-age cohorts.

## 2. Materials and methods

### 2.1. Data sources and study design

This study is a secondary ecological analysis of publicly available GBD 2023 data. We examined aggregated national-level estimates of HIV/AIDS burden without direct involvement of human participants. We used the Global Burden of Disease Study 2023 (GBD 2023) as our data source; estimates are publicly available at https://vizhub.healthdata.org/gbd-results/. The data sources and modelling methods for GBD 2021 have been described in detail elsewhere [10–12]; that round provided a comprehensive assessment of disease, injury and risk-factor burdens by age, sex and year for 1990–2021 across 204 countries and territories, covering 371 diseases and injuries and 88 risk factors [10]. The 2023 iteration of GBD, however, remains largely under-utilised in the current literature.

We selected the following parameters: Location: China; Cause: HIV/AIDS; Metric: Number, Rate, and Percent; Age: 15–19, 20–24, …, 45–49 years (five-year age groups); Sex: Male, Female, Both; Year: 1990–2023; Measure: Incidence, Prevalence, Mortality, and DALYs. Data were downloaded as CSV files in December 2025 and imported into R4.3.2 for analysis. Age-standardised rates were then calculated by applying the GBD 2023 world-standard population weights to these crude rates, with population denominators taken from the GBD 2023 population module. IHME generated these estimates using the Estimation and Projection Package 2023/Spectrum 6.3 (EPP 2023/Spectrum 6.3), synthesising HIV sentinel surveillance, vital registration and antiretroviral-therapy programme data; uncertainty was propagated through 500 Monte-Carlo draws. We used posterior means and their 95% uncertainty intervals to derive standard errors for joinpoint regression (SE = [upper − lower]/3.92) as required by Joinpoint 5.2 software; ARIMA models used posterior mean estimates as point inputs without uncertainty weighting, with prediction intervals derived from the model's error variance.. All 95% UIs are those produced by the GBD 2023 Bayesian meta-regression platform; no additional Bayesian modelling was performed by the authors.

### 2.2. Statistical methods

Four age-standardised indicators were calculated: (i) incidence rate (ASIR), new HIV cases per 100,000 population; (ii) prevalence rate (ASPR), existing HIV cases per 100,000; (iii) mortality rate (ASMR), HIV-related deaths per 100,000; and (iv) disability-adjusted life-year rate (ASDR), DALYs attributable to HIV/AIDS per 100,000. All rates used the GBD 2023 world-standard population and were plotted over time to describe and visualise secular trends of HIV/AIDS burden in China.



Joinpoint regression (version 5.2.0, National Cancer Institute, USA) was applied to the 1990–2023 age-standardised rate series to detect significant inflection points. The default Monte Carlo permutation test (4,499 permutations, α = 0.05) was used to select the optimal number and location of joinpoints. Each segment's trend was quantified by the annual percentage change (APC) and the overall average annual percentage change (AAPC) with 95% confidence intervals; an AAPC > 0 indicates an upward trend, whereas an AAPC < 0 indicates a downward trend [13].

To assess the robustness of joinpoint locations and trend estimates, two additional model sets were fitted. First, the optimal number of joinpoints was re-evaluated using the Bayesian Information Criterion (BIC), and the direction and significance of the resulting AAPC were compared with those of the main model. Second, a time-window sensitivity analysis was performed by restricting the series to 2000–2023 and refitting the model; the direction of the latest-segment APC, 95% confidence interval overlap and ΔAAPC between the two windows were examined. Overlapping 95% CIs or a ΔAAPC < 5 percentage points were considered indicative of robust findings.

Absolute counts and crude rates per 100,000 by sex and 5-year age group were extracted for China. Population pyramids were constructed with the ggplot2 package in R version 4.3.2 (R Core Team, 2023): age groups on the y-axis, absolute cases on the x-axis, with males displayed leftward and females rightward. Crude-rate trends were plotted as line graphs (age group on x-axis; rate on y-axis; colour-coded by sex). Rate ratios (male rate / female rate) were calculated with spreadsheet software (WPS Office) to quantify sex disparities across age groups.

ARIMA models were fitted with R 4.3.2 and the forecast package. Because sex-specific ASIR series failed white-noise diagnostics for 1990–2023 (Ljung-Box p < 0.05), male and female ASIR were separately modelled on 2000–2023 data, while ASPR, ASMR and ASDR used the full 1990–2023 series. This approach balances statistical model requirements with maximum use of available data; joinpoint sensitivity analyses confirmed robustness of trend estimates to alternative model specifications and time-window restrictions (Supplementary S2–S3 Tables). Model diagnostics included the Ljung-Box test for residual white noise and inspection of residual autocorrelation functions. All final models showed no significant residual autocorrelation (Ljung-Box p > 0.05, S1 Table) and were selected based on the lowest corrected Akaike information criterion (AICc). All series were kept on the natural scale; orders p ≤ 3 and q ≤ 3 were compared and the optimal specification retained. Ninety-five percent prediction intervals were derived from the ARIMA model's error variance using the forecast package's analytical formulae; these intervals reflect model-based uncertainty and do not incorporate additional uncertainty from the GBD input estimates.

HIV/AIDS attributable deaths and DALYs were analysed for four behavioral risk factors: unsafe sex, drug use, intimate-partner violence, and sexual violence against children and bullying. Factor selection followed the GBD 2021 Scientific Council criteria [10] and Lu et al.'s global analysis [6]. Within the GBD three-tier hierarchy we evaluated all Level 2 behavioral risks available for China; sexual violence against children and bullying was excluded because its 95% uncertainty interval included negative values (lower bound < 0), indicating unstable estimates inconsistent with the counterfactual causal framework. Environmental and metabolic risks were also excluded because no attributable burden estimates were available for HIV/AIDS in China. We retained unsafe sex, drug use, and intimate-partner violence as the three risk factors with robust, non-crossing attributable burden estimates for China. We extracted the population attributable fractions (PAFs) — (attributable deaths or DALYs ÷ total deaths or DALYs) × 100% — directly from the GBD 2023 results tool; no additional calculations were performed. Uncertainty was propagated through the 1 000 posterior draws supplied by GHDx.

Ethics statement: This study used publicly available, de-identified data from the Global Burden of Disease Study 2023. Ethics approval and informed consent were not required.

## 3. Results

### 3.1. Disease burden of HIV/AIDS among reproductive-age adults in China (1990–2023)

Between 1990 and 2023, both absolute numbers and age-standardised rates of HIV/AIDS among reproductive-age adults (15–49 years) in China rose continuously (Table 1). Incident cases increased from 5,475–21,242 (total change +288%;



**Table 1. HIV/AIDS Incidence and Prevalence among Reproductive-Age Populations in China, 1990–2023.**

| measure | sex | metric | 1990 (95%UI) | 2023 (95%UI) | Change(%) | AAPC (95%CI) | P |
|---|---|---|---|---|---|---|---|
| Incidence | both | Number(n) | 5,475 (3,253−9,390) | 21,242 (11,281−36,605) | 287.98 | 4.07 (2.97-5.17) | <0.001 |
| | | Rate(per 100,000) | 0.82 (0.49-1.40) | 3.29 (1.75-5.67) | 301.22 | 4.18 (3.10-5.28) | <0.001 |
| | | ASIR(per 100,000) | 0.82 (0.46-1.44) | 3.29 (1.59-5.67) | 301.22 | 4.18 (3.10-5.28) | <0.001 |
| | male | Number(n) | 3,635 (2,125−6,292) | 17,112 (9,231−29,131) | 370.76 | 4.79 (3.60-5.98) | <0.001 |
| | | Rate(per 100,000) | 1.05 (0.61-1.82) | 5.08 (2.74-8.64) | 383.81 | 4.87 (3.67-6.07) | <0.001 |
| | | ASIR(per 100,000) | 1.05 (0.61-1.86) | 5.08 (2.49-8.75) | 383.81 | 4.87 (3.68-6.07) | <0.001 |
| | female | Number(n) | 1,840 (1,100−3,186) | 4,131 (2,050−7,376) | 124.51 | 2.39 (1.41-3.37) | <0.001 |
| | | Rate(per 100,000) | 0.57 (0.34-0.99) | 1.34 (0.66-2.39) | 135.09 | 2.56 (1.53-3.60) | <0.001 |
| | | ASIR(per 100,000) | 0.57 (0.33-0.99) | 1.34 (0.62-2.37) | 135.09 | 2.56 (1.58-3.56) | <0.001 |
| Prevalence | both | Number(n) | 32,372 (15,161−44,982) | 273,064 (134,284-536,669) | 743.52 | 6.66 (6.51-6.82) | <0.001 |
| | | Rate(per 100,000) | 4.84 (2.27-6.73) | 42.29 (20.80-83.12) | 773.76 | 6.81 (6.65-6.97) | <0.001 |
| | | ASPR(per 100,000) | 4.84 (2.28-6.84) | 42.29 (20.77-82.95) | 773.76 | 6.81 (6.65-6.97) | <0.001 |
| | male | Number(n) | 21,557 (9,741−30,245) | 211,111 (103,546-410,171) | 879.32 | 7.16 (6.99-7.33) | <0.001 |
| | | Rate(per 100,000) | 6.24 (2.82-8.75) | 62.62 (30.72-121.67) | 903.53 | 7.27 (7.08-7.46) | <0.001 |
| | | ASPR(per 100,000) | 6.24 (2.83-8.90) | 62.62 (30.85-121.69) | 903.53 | 7.27 (7.08-7.46) | <0.001 |
| | female | Number(n) | 10,815 (5,419−14,717) | 61,936 (30,646−123,644) | 472.69 | 5.39 (5.20-5.58) | <0.001 |
| | | Rate(per 100,000) | 3.35 (1.68-4.56) | 20.08 (9.93-40.08) | 499.40 | 5.57 (5.42-5.72) | <0.001 |
| | | ASPR(per 100,000) | 3.35 (1.70-4.62) | 20.08 (9.80-40.46) | 499.40 | 5.57 (5.42-5.72) | <0.001 |

AAPC 4.07%, 95% CI 2.97–5.17), while ASIR grew from 0.82 to 3.29 per 100,000 (+301%; AAPC 4.18%, 3.10–5.28). Prevalent cases climbed from 32,372–273,064 (+744%; AAPC 6.66%, 6.51–6.82) and ASPR from 4.84 to 42.29 per 100,000 (+774%; AAPC 6.81%, 6.65–6.97). Annual deaths rose from 890 to 14,335 (+1,511%; AAPC 8.31%, 7.24–9.40) and ASMR from 0.13 to 2.24 per 100,000 (+1,623%; AAPC 8.53%, 7.49–9.58). DALYs increased from 53,242–784,921 (+1,374%; AAPC 8.32%, 7.43–9.22) and the age-standardised DALY rate from 7.96 to 122.48 per 100,000 (+1 439%; AAPC 8.38%, 7.43–9.34). All trends were upward ($p < 0.001$). These large percentage increases reflect the very low baseline rates in 1990, when the HIV epidemic was in its early stages in China. Notably, age-standardised incidence peaked around 2003–2004 and has since declined or plateaued (Fig 1A), indicating that the epidemic is no longer in a phase of rapid expansion. The continued rises in prevalence, mortality, and DALYs reflect improved survival on antiretroviral therapy and population ageing rather than escalating transmission. Throughout 1990–2023, both absolute numbers



**Fig 1. Joinpoint regression of age-standardised rates (ASR) in Chinese reproductive-age population by sex, 1990–2023: (A) APC of ASIR; (B) APC of ASPR; (C) APC of ASMR; (D) APC of ASDR.** Dots denote significant APC (P < 0.05); vertical bars indicate 95% confidence intervals.

and age-standardised rates remained consistently higher in males, and all upward trends were steeper in males than in females (Tables 1,2).

### 3.2. Sex-specific joinpoint regression analysis of HIV/AIDS burden among reproductive-age adults in China

Between 1990 and 2023, all four HIV/AIDS metrics among Chinese adults aged 15–49 years increased continuously, with consistently steeper trends in males than in females (Tables 1,2, Fig 1).

ΔAAPC (males − females): ASIR +2.30 pp ($p = 0.004$), ASPR +1.70 pp ($p < 0.001$), ASMR +2.02 pp ($p = 0.013$), ASDR +2.08 pp ($p = 0.006$).

Segment-wise analyses revealed two rapid-increase periods (1997–2001 and 2013–2023) for incidence; prevalence growth slowed after 2014; mortality and DALY rates plateaued during 2005–2013 and 2020–2023, but rose sharply in all other segments (Fig 1).

Sensitivity analyses showed that alternative BIC-based models and the 2000–2023 window produced ΔAAPC < 0.5 pp and overlapping 95% CIs, confirming robustness (S2–S3 Tables); however, excluding 1990–1999 reversed the incidence direction (ΔAPC = −4.04%) while prevalence, mortality and DALY trends remained stable.

**Table 2. HIV/AIDS Deaths and DALYs among Reproductive-Age Populations in China, 1990–2023.**

| measure | sex | metric | 1990 (95%UI) | 2023 (95%UI) | Change(%) | AAPC (95%CI) | P |
|---|---|---|---|---|---|---|---|
| Deaths | both | Number(n) | 890 (26−1,640) | 14,335 (10,079−18,641) | 1,510.67 | 8.31 (7.24-9.40) | <0.001 |
| | | Rate(per 100,000) | 0.13 (0.00-0.25) | 2.22 (1.56-2.89) | 1,607.69 | 8.48 (7.50-9.46) | <0.001 |
| | | ASMR(per100,000) | 0.13 (0.00-0.26) | 2.24 (1.48-3.21) | 1,623.08 | 8.53 (7.49-9.58) | <0.001 |
| | male | Number(n) | 574 (17−1,102) | 11,266 (7,779−14,913) | 1,862.72 | 9.08 (8.06-10.12) | <0.001 |
| | | Rate(per 100,000) | 0.17 (0.00-0.32) | 3.34 (2.31-4.42) | 1,864.71 | 9.23 (8.30-10.17) | <0.001 |
| | | ASMR(per100,000) | 0.17 (0.00-0.34) | 3.37 (2.12-5.00) | 1,882.35 | 9.28 (8.31-10.25) | <0.001 |
| | female | Number(n) | 317 (10-575) | 3,069 (2,141−3,965) | 868.14 | 6.70 (5.47-7.94) | <0.001 |
| | | Rate(per 100,000) | 0.10 (0.01-0.18) | 0.99 (0.69-1.28) | 890.00 | 6.89 (5.76-8.03) | <0.001 |
| | | ASMR(per100,000) | 0.10 (0.01-0.19) | 1.00 (0.62-1.52) | 900.00 | 7.26 (6.01-8.52) | <0.001 |
| DALYs | both | Number(n) | 53,242 (4,476−93,036) | 784,921 (554,735−1,033,893) | 1,374.25 | 8.32 (7.43-9.22) | <0.001 |
| | | Rate(per 100,000) | 7.96 (0.67-13.91) | 121.57 (85.92-160.14) | 1,427.26 | 8.22 (7.26-9.18) | <0.001 |
| | | ASDR(per 100,000) | 7.96 (0.68-14.69) | 122.48 (81.44-174.95) | 1,438.69 | 8.38 (7.43-9.34) | <0.001 |
| | male | Number(n) | 34,041 (2,800−62,493) | 614,432 (427,582-817,981) | 1,704.98 | 9.08 (8.19-9.98) | <0.001 |
| | | Rate(per 100,000) | 9.85 (0.81-18.08) | 182.26 (126.84-242.65) | 1,750.36 | 8.97 (8.05-9.90) | <0.001 |
| | | ASDR(per 100,000) | 9.85 (0.80-18.98) | 183.60 (117.23-270.11) | 1,763.96 | 9.02 (8.07-9.98) | <0.001 |
| | female | Number(n) | 19,201 (1,640−33,373) | 170,489 (118,567−225,042) | 787.92 | 6.43 (5.24-7.63) | <0.001 |
| | | Rate(per 100,000) | 5.94 (0.51-10.33) | 55.26 (38.43-72.94) | 830.30 | 6.62 (5.54-7.71) | <0.001 |
| | | ASDR(per 100,000) | 5.94x (0.52-10.87) | 55.70 (34.93-83.53) | 837.71 | 6.94 (5.81-8.08) | <0.001 |

### 3.3. Sex- and age-specific distribution of absolute HIV/AIDS burden in China's reproductive-age population, 2023

Across all age groups in 2023, males carried a substantially higher HIV/AIDS burden than females in terms of absolute case numbers (Fig 2). Incident cases peaked later in males at 30–34 years (approximately 20,000 cases) compared with females at 25–29 years (approximately 5,000 cases), representing a fourfold difference (Fig 2A). Prevalence counts remained elevated from 30–49 years in both sexes, but male numbers rose dramatically after age 35, peaking at >150,000 cases in the 45–49-year band, whereas female prevalence peaked at approximately 30,000 cases in the 35–39-year group (Fig 2B).



**Fig 2. Age–sex pyramid of HIV/AIDS burden, China, 2023: (A) incident cases; (B) prevalent cases; (C) deaths; (D) DALYs.** Males (right) and females (left) are plotted as absolute counts.

Mortality and DALYs both demonstrated unimodal patterns concentrated in middle age. For males, absolute deaths peaked at 45–49 years (nearly 10,000 deaths), while females reached their maximum in the same 45–49-year group (approximately 2,000 deaths) (Fig 2C). Similarly, male DALYs peaked at 30,000 person-years in the 45–49-year band, approximately four times the female peak of 7,000 person-years (Fig 2D), with detailed rate patterns presented in section 3.4.

### 3.4. Sex- and age-specific patterns of HIV/AIDS rates in China's reproductive-age population

Age-specific crude rates demonstrated pronounced male predominance across all indicators (Fig 3). Male incidence peaked at 25–29 years (~25 per 100,000), approximately 2.5 times the female peak at 20–24 years, with the sex ratio expanding to 4:1 in 35–39-year-olds (Fig 3A, S4 Table). Male prevalence culminated at 40–44 years (~200 per 100,000), generating a 3:1 male-to-female ratio during 35–49 years (Fig 3B). Mortality and DALYs exhibited comparable patterns, with male rates peaking at 45–49 years (~40 per 100,000 deaths; ~1,500 per 100,000 DALYs) and exceeding female rates by >3.5-fold, while female rates stabilized after age 30 (Fig 3C–D). This divergence—escalating male rates through midlife versus plateauing female rates—produced a progressively widening sex disparity with advancing age.

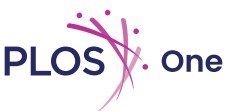

**Fig 3. Age-specific crude rates by sex, China, 2023: (A) incidence; (B) prevalence; (C) mortality; (D) DALYs.** Rates are per 100,000 population.

### 3.5. Attributable analysis of HIV/AIDS mortality and DALYs among reproductive-age individuals in China

This study systematically evaluated the contribution of intimate partner violence, drug use, and unsafe sexual practices to HIV/AIDS-related mortality and disability-adjusted life years (DALYs) in the reproductive-age population (Fig 4A, 5A). Annual PAF estimates from 1990 to 2023 (Supplementary S1–S6 Fig) demonstrated that unsafe sexual practices constituted the predominant risk factor, maintaining a high attributable burden among individuals aged 15–29 years (Fig 4D, 5D). Drug use represented the second major contributor, with an age-gradient effect leading to peak burden in the 40–49 age group (Fig 4C, 5C). Intimate partner violence demonstrated distinct sex- and age-specific patterns, with burden peaking in women aged 30–39 years in 1990 and transitioning to those aged 40–49 years by 2023 (Fig 4B, 5B).

### 3.6. Projection of HIV/AIDS burden among Chinese women of reproductive age, 2024–2035

Using GBD 2023, we fitted ARIMA models to ASIR (2000–2023) and to ASPR, ASDR and ASMR (1990–2023) (Methods in Appendix). Under the "current-control-effort unchanged" scenario, the predicted trajectory to 2035 shows a clear sex gap among people aged 15–49 years in China.



**Fig 4. Population attributable fractions (PAFs) of HIV/AIDS deaths by behavioral risk factors among reproductive-age adults in China, 1990 and 2023. (A)** Total PAFs by risk factor and sex. **(B–D)** Age-specific PAFs for (B) intimate partner violence (females), (C) drug use, and (D) unsafe sex. Dynamic temporal trends (1990–2023) for deaths are shown in Supplementary Videos S1–S3. PAFs and 95% uncertainty intervals were obtained directly from GBD 2023. PAF, population attributable fraction.

**3.6.1. Men.** Incidence and prevalence have plateaued: ASIR will remain ≈ 5.0 and ASPR ≈ 63.5 per 100,000 by 2035 (Fig 6A-B). ASMR falls from 3.1 in 2024 to 2.6 per 100,000 in 2035; ASDR declines in parallel to 145 per 100,000 by 2035 (Fig 6C-D).

**3.6.2. Women.** All four indicators keep rising:ASIR: 1.4→1.7 per 100,000 (+21%); ASPR: 20.2→26.3 per 100,000 (+30%); ASMR: 0.93→1.20 per 100,000 (+29%); ASDR: 50.7→62.5 per 100,000 (+23%) (Fig 6A-D).

In sum, men show stable incidence and declining mortality, whereas women exhibit sustained increases across all metrics, yielding a marked sex difference (S3 Table).

## 4. Discussion

This study represents an updated comprehensive assessment of HIV/AIDS burden specifically among reproductive-age adults in China, leveraging the most recent GBD 2023 data, and focusing on a critical demographic for epidemic control and economic productivity. Four key findings emerge: (i) age-standardised incidence has plateaued but prevalence-related mortality continues rising, reflecting the transition from acute infection emergency to chronic disease burden [1,2]; (ii) males bear disproportionately higher burden across all metrics, consistent with higher reported rates of MSM



**Fig 5. Population attributable fractions (PAFs) of HIV/AIDS DALYs by behavioral risk factors among reproductive-age adults in China, 1990 and 2023. (A)** Total PAFs by risk factor and sex. **(B–D)** Age-specific PAFs for (B) intimate partner violence (females), (C) drug use, and (D) unsafe sex. Dynamic temporal trends (1990–2023) for DALYs are shown in Supplementary S4–S6 Fig. See Fig 4 legend for additional details. DALYs, disability-adjusted life years.

transmission and drug use; (iii) unsafe sex and drug use account for the majority of attributable deaths, identifying clear intervention priorities; and (iv) sex-specific divergence will persist through 2035 without intensified, gender-differentiated strategies. While previous analyses examined broader age ranges, our focus on the reproductive-age population (15–49 years) provides critical evidence for this economically and demographically critical segment [8]. These findings offer timely evidence for China's newly launched National HIV/AIDS Action Plan (2024–2030) and measurable targets for SDG 3.3 monitoring, while this analytical approach is readily transferable to other middle-income countries facing similar epidemiological transitions [3,14].

Attributable risk analysis identifies actionable targets for infectious disease control. Unsafe sex accounted for 70% of HIV/AIDS mortality in 2023. This proportion is consistent with global patterns, where sexual transmission dominates generalized epidemics [6,7]. The GBD 2021 risk factor hierarchy provides a standardized framework. It enables attributable burden estimation across diverse settings [10]. However, China's distinct feature is the substantial contribution of drug use, particularly among middle-aged males. This pattern reflects the historical injection drug use epidemic and ongoing transmission networks [15]. Intimate-partner violence, contributes smaller absolute fractions. Nonetheless, it represents a



**Fig 6. Forecasted age-standardised rates by sex among China's reproductive-age population, 2024–2035: (A) ASIR; (B) ASPR; (C) ASMR; (D) ASDR.** Bands represent 95% uncertainty intervals from ARIMA models fitted to GBD 2023 data.

preventable, gender-specific risk factor concentrated among women aged 30–49 years [16]. These population attributable fractions provide clear intervention priorities. Intensifying condom promotion and harm reduction for drug users could yield disproportionate mortality reductions. Our supplementary analyses (S1–S6 Fig) demonstrate dynamic trends across age and time. Such data enable tailored, life-stage-specific prevention strategies.

The temporal trajectories we observed parallel China's evolving HIV policy landscape. The steep rise in incidence from 1990 to 2004 reflects limited public knowledge and scarce prevention services in the early epidemic phase [1]. occurring from an extremely low baseline when HIV was first emerging in China. The 2003 creation of the National AIDS Prevention and Control Fund and 2004 Regulations providing free ART reversed this trend, demonstrating the impact of sustained government investment [17]. While the large percentage increases in cumulative burden between 1990 and 2023 appear substantial, they primarily reflect this low baseline and early expansion rather than recent epidemic growth. The continued rise in prevalence-related mortality and DALYs despite plateauing incidence underscores a critical challenge: the growing chronic-care demand from an ageing, treatment-experienced population [2]. This pattern contrasts with sub-Saharan Africa, where scaled-up treatment has reduced prevalence, and highlights the distinct trajectory of concentrated epidemics in middle-income settings [18]. The growing political priority for sexually

transmitted infection control further underscores the need to shift from emergency response to integrated chronic disease management [4].

The pronounced male predominance across all burden metrics, with 2- to 4-fold higher rates than females, reflects the dominant MSM and drug use transmission dynamics in China [19,20]. This contrasts with sub-Saharan Africa and other generalized epidemic settings where adolescent and young adult women carry disproportionate burden due to biological susceptibility and gender-based power imbalances [16,21]. The male predominance in China may indicate that current test-and-treat and PrEP scale-up strategies are reaching insufficient coverage in key populations. Peak incidence at 20–24 years among men, attributable to higher partner turnover and low condom consistency, indicates that young adult MSM remain a critical intervention target [22]. Conversely, plateauing female incidence demonstrates prevention successes among women that should be maintained through continued antenatal screening and partner notification programs.

Our ARIMA projections to 2035 reveal a concerning divergence. While male incidence and mortality are stabilising, all four indicators continue rising among women. These trends include age-standardised incidence, prevalence, mortality, and DALYs rates. Under current control efforts, China will likely fall short of key UNAIDS targets. These include the 95-95-95 cascade goals and the 90% reduction target for new infections and deaths by 2030 [14]. The 21–30% projected increases in female metrics are particularly alarming, threatening overall epidemic control. Achieving elimination targets requires intensified, gender-differentiated strategies. These include scaling up community-based testing and online outreach for MSM, sustaining harm reduction for drug users, enhancing PrEP accessibility for women, and dismantling stigma that impedes testing and treatment across all populations [23,24]. School-based interventions for adolescents remain particularly important. This is essential given the peak incidence in the 20–24 age group [25].

The combination of Joinpoint regression and ARIMA modeling represents a robust analytical framework for infectious disease surveillance and forecasting. This analytical approach has been widely validated in global HIV burden assessments [7,26]. The GBD study's standardized methodology. These enable cross-national comparability and policy-relevant quantification [12]. Joinpoint analysis identifies significant trend inflection points. This capability allows precise assessment of policy impacts. ARIMA models complement this by providing validated short-to-medium term projections. Such projections are essential for resource planning. This approach is readily transferable to other regional HIV programs. It is also adaptable for emerging infectious diseases. These features collectively support evidence-based public health decision-making in diverse settings.

This study has several limitations. First, these interpretations are based on ecological associations and should be considered hypothesis-generating rather than definitive causal inferences; individual-level data on transmission routes and risk behaviors were not available in the GBD aggregated estimates. Second, GBD 2023 data estimates incorporate surveillance data subject to underreporting biases and time lag. Underreporting may occur due to stigma, limited testing coverage, or diagnostic gaps—particularly in the early epidemic period (1990s)—while the inherent data release lag (public release October 2025), potentially limiting capture of the most recent epidemic dynamics, including post-COVID-19 recovery patterns. Although GBD applies spatiotemporal Gaussian process regression and Cause of Death Ensemble modelling (CODEm) to adjust for these biases, residual uncertainty persists and is reflected in our reported 95% uncertainty intervals. Third, unaccounted risk factors and contextual variables—such as concurrent sexually transmitted infections and mobility patterns—may introduce residual confounding into burden estimates. Fourth, ARIMA projections assume continuity of historical trends and do not account for external shocks such as policy changes or behavioral shifts. Prediction intervals substantially widen with forecast horizon, so we have limited primary projections to 2030–2035 to minimize long-term extrapolation error. Finally, province-level heterogeneity was not examined. This aggregation may mask substantial regional variations, such as distinct epidemiological profiles between southwestern and eastern coastal provinces. Subnational analyses will be important for targeted resource allocation and are the focus of subsequent work.

China's HIV epidemic among reproductive-age adults has transitioned from an acute infection emergency to a chronic disease burden. This burden shows distinct sex-specific profiles. Age-standardised incidence is stabilising overall. However, prevalence-related mortality and disability will keep rising without intensified intervention. This trend is particularly pronounced among men. Unsafe sex and drug use emerge as priority targets for attributable mortality reduction. These findings suggest the value of integrating HIV management into routine infectious disease surveillance and chronic care systems. with consideration of Gender-differentiated strategies to avert long-term disability and mortality [23,24]. The projected 21–30% increases in female burden indicators highlight a need for attention. Expanding PrEP accessibility and harm reduction services for women could help address these trends. Maintaining MSM-focused strategies remains important, given the plateau in male incidence. These actionable metrics provide evidence for China's National HIV/AIDS Action Plan (2024–2030). They also offer measurable targets for SDG 3.3 monitoring. This analytical framework may be applicable to other middle-income countries facing similar epidemiological transitions. It supports evidence-based public health decision-making in diverse settings.

## Supporting information

**S1 Table. Ljung-Box test results for ARIMA models.**
(XLSX)

**S2 Table. Joinpoint sensitivity analysis I (BIC-based model selection).**
(XLSX)

**S3 Table. Joinpoint sensitivity analysis II (time window 2000–2023).**
(XLSX)

**S4 Table. Sex-specific rate ratios (crude, M/F) by age group, China, 2023.**
(XLSX)

**S1 Fig. Temporal trends of PAFs for HIV/AIDS deaths attributable to drug use among reproductive-age adults in China, 1990–2023.**
(GIF)

**S2 Fig. Temporal trends of PAFs for HIV/AIDS deaths attributable to unsafe sex among reproductive-age adults in China, 1990–2023.**
(GIF)

**S3 Fig. Temporal trends of PAFs for HIV/AIDS deaths attributable to intimate partner violence among reproductive-age females in China, 1990–2023.**
(GIF)

**S4 Fig. Temporal trends of PAFs for HIV/AIDS DALYs attributable to drug use among reproductive-age adults in China, 1990–2023.**
(GIF)

**S5 Fig. Temporal trends of PAFs for HIV/AIDS DALYs attributable to unsafe sex among reproductive-age adults in China, 1990–2023.**
(GIF)

**S6 Fig. Temporal trends of PAFs for HIV/AIDS DALYs attributable to intimate partner violence among reproductive-age females in China, 1990–2023.**
(GIF)



## Acknowledgments

This study gratefully acknowledges the support of the GBD database platform and the Shanghai Jiao Tong University Library.

## Author contributions

**Conceptualization:** Huibo Yan.

**Data curation:** Huibo Yan.

**Formal analysis:** Huibo Yan.

**Investigation:** Huibo Yan.

**Methodology:** Huibo Yan.

**Project administration:** Huibo Yan.

**Software:** Huibo Yan.

**Visualization:** Huibo Yan.

**Writing – original draft:** Huibo Yan.

**Writing – review & editing:** Huibo Yan.

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
