## [Decision Letter · Decision Letter 0]

19 Mar 2026

PONE-D-26-08383HIV/AIDS burden, attributable risk factors, and projections among reproductive-age adults in China, 1990–2035: a GBD 2023 analysisPLOS One

Dear Dr. Yan,

Thank you for submitting your manuscript to PLOS ONE. After careful consideration, we feel that it has merit but does not fully meet PLOS ONE’s publication criteria as it currently stands. Therefore, we invite you to submit a revised version of the manuscript that addresses the points raised during the review process.

We look forward to receiving your revised manuscript.

Kind regards,

Ahmad Ghashghaee

Academic Editor

PLOS One

Journal Requirements:

2. Please amend either the abstract on the online submission form (via Edit Submission) or the abstract in the manuscript so that they are identical.

**Additional Editor Comments:**

The framing of the research question lacks sufficient specificity and should more clearly define the population, exposure (if applicable), and outcome measures to enhance conceptual clarity.

The Introduction would be strengthened by incorporating a more critical synthesis of recent high-impact literature, rather than relying primarily on descriptive background information.

The justification for the selected time frame should be explicitly provided, particularly if it aligns with major global health transitions or data availability constraints.

The Methods section should more clearly describe how data harmonization across heterogeneous sources was achieved, especially if multiple surveillance or registry systems were used.

The manuscript would benefit from greater transparency regarding inclusion and exclusion criteria at the dataset level, particularly if certain regions or populations were systematically omitted.

The authors should clarify how missing data were handled, including whether imputation techniques were applied and how they may influence the robustness of estimates.

The statistical modeling approach requires more detailed justification, including the assumptions underlying the selected model(s) and their appropriateness for the data structure.

The manuscript should explicitly address whether model diagnostics and goodness-of-fit assessments were conducted and how these informed model selection.

The presentation of uncertainty should be strengthened by explaining how uncertainty intervals were propagated through different stages of analysis.

The Results section would benefit from improved structuring to clearly separate primary findings from exploratory or secondary analyses.

Several results are presented descriptively without sufficient interpretation; the manuscript should provide analytical insights into observed patterns, rather than reiterating numerical outputs.

The manuscript would benefit from clearer identification of key drivers of change, including demographic shifts, behavioral factors, or policy interventions.

The Discussion should more rigorously engage with alternative interpretations of the findings, particularly where causal inference cannot be established.

The authors should more explicitly acknowledge the potential for systematic biases in data sources, including reporting bias, surveillance bias, and measurement error.

Greater attention should be given to external validity, particularly whether findings can be generalized beyond the studied populations or settings.

The manuscript would benefit from a more detailed exploration of equity dimensions, including disparities across socioeconomic strata, geographic regions, or vulnerable populations.

The policy implications section should be more tightly aligned with the empirical findings and avoid overgeneralization or normative statements not directly supported by the data.

The limitations section should be expanded to include methodological constraints related to data comparability and temporal consistency.

The manuscript should provide a clearer distinction between statistical significance and practical or public health significance.

Figures and tables should be revised to enhance clarity, including standardized formatting, clearer labeling, and improved visual hierarchy.

The manuscript would benefit from reducing redundancy between the Results and Discussion sections to improve overall coherence.

A brief statement on data accessibility and reproducibility, including availability of analytical code or datasets, would enhance transparency.

The conclusion should be more concise and focused on the most critical findings, avoiding repetition of background information.

Overall, the manuscript would benefit from a more rigorous alignment between research objectives, analytical methods, and interpretative claims to strengthen its scientific contribution.

Reviewers' comments:

Reviewer's Responses to Questions

**Comments to the Author**

1. Is the manuscript technically sound, and do the data support the conclusions?

Reviewer #1: Yes

Reviewer #2: Yes

2. Has the statistical analysis been performed appropriately and rigorously? 

Reviewer #1: Yes

Reviewer #2: Yes

3. Have the authors made all data underlying the findings in their manuscript fully available?

Reviewer #1: Yes

Reviewer #2: Yes

4. Is the manuscript presented in an intelligible fashion and written in standard English?

Reviewer #1: Yes

Reviewer #2: Yes

5. Review Comments to the Author

Reviewer #1: This manuscript analyzes the burden, attributable risk factors, and projected trends of HIV/AIDS among reproductive-age adults (15–49 years) in China using Global Burden of Disease (GBD) 2023 data, applying Joinpoint regression and ARIMA forecasting models. The topic is relevant for epidemiological surveillance and health policy planning, and the use of standardized GBD estimates allows for long-term trend assessment.

Overall, the study provides a useful descriptive overview of HIV burden trends in China. However, several issues related to clarity, methodological description, interpretation of results, and manuscript structure should be addressed before the manuscript can be considered for publication. In particular, the introduction does not clearly articulate the knowledge gap, parts of the methods section require clarification, and some interpretations in the discussion appear overly strong given the ecological nature of the data.

Major comments

1. Introduction – clarity of the research gap

The introduction provides a broad overview of HIV/AIDS epidemiology globally and in China, but the specific research gap addressed by this study is not clearly articulated.

While the manuscript states that this is the “first comprehensive analysis” of HIV burden in reproductive-age adults using GBD 2023 data, this claim should be better justified or moderated. Previous studies using earlier GBD datasets have already examined HIV burden trends in China and globally.

The authors should more clearly explain:

• Why focusing specifically on the 15–49 age group provides additional insight beyond existing studies.

• What new contribution the use of GBD 2023 data provides compared with GBD 2021 or earlier analyses.

• How this study advances the current literature on HIV epidemiology in China.

2. Methods – clarification of study design

The study relies entirely on secondary data from the Global Burden of Disease database. However, the manuscript does not explicitly state the study design.

The authors should clarify early in the methods section that this is a secondary ecological analysis of publicly available GBD data.

Additionally, more detail should be provided on:

• The exact data extraction procedures from the GBD platform.

• Whether uncertainty intervals from the GBD estimates were incorporated into trend analyses.

• The rationale for restricting the ARIMA incidence model to the 2000–2023 time series.

3. Selection of risk factors

The study examines three risk factors: unsafe sex, drug use, and intimate partner violence. However, the process by which these factors were selected requires further clarification.

The authors state that the selection followed the GBD risk factor hierarchy, but it remains unclear:

• Whether other risk factors were evaluated and subsequently excluded.

• Whether the selected risk factors represent the full set of attributable factors available in the GBD framework for HIV/AIDS.

This should be clarified to improve methodological transparency.

4. Interpretation of large percentage increases

The manuscript reports very large percentage increases in incidence, prevalence, mortality, and DALYs between 1990 and 2023.

While mathematically correct, these large percentage increases may largely reflect very low baseline rates in the early years of the epidemic. This point should be explicitly acknowledged in the Results or Discussion sections to avoid misleading interpretation.

5. Interpretation of causal mechanisms

Some statements in the discussion imply causal explanations (for example, linking male predominance directly to MSM transmission patterns and drug use). While these explanations are plausible, the analysis is based on aggregated ecological data, and therefore causal inference should be interpreted cautiously.

The authors should moderate language suggesting causal mechanisms and clearly distinguish between observed associations and hypothesized explanations.

6. Forecasting analysis

The ARIMA-based projections to 2035 are a valuable component of the study. However, additional clarification is needed regarding:

• Model diagnostics beyond the Ljung–Box test.

• How uncertainty intervals were propagated into the projections.

• The limitations of ARIMA models for long-term epidemiological forecasting.

These points should be discussed more explicitly.

7. Redundancy in results and discussion

Several sections of the manuscript contain repetitive descriptions of the same findings. For example, the male predominance in HIV burden and the transition toward chronic disease burden are mentioned multiple times across the Results and Discussion sections.

The manuscript would benefit from condensing repetitive sections and improving narrative flow.

Minor comments

1. There is a typographical error in the section heading:

“Intoduction” should be corrected to “Introduction”.

2. The manuscript should standardize the formatting of units, particularly:

o “per 100 000”

o “per 100,000”

3. Some sentences are excessively long and difficult to follow. Language editing by a fluent English speaker would improve readability.

4. The legends for several figures are overly technical and could be simplified to improve accessibility for readers.

5. Table 1 is very dense and may benefit from simplification or splitting into separate tables for incidence/prevalence and mortality/DALYs.

Comments on figures and tables

The figures are informative but could be improved for clarity.

• Figure 1: Joinpoint plots are useful, but the legend is complex and may be difficult for readers unfamiliar with Joinpoint methodology.

• Figure 2: The population pyramid visualization is interesting, but the scaling makes visual comparison between sexes somewhat difficult.

• Table 1: The table contains redundant metrics (e.g., rate and ASIR). Consider simplifying the presentation to improve readability.

Limitations

The limitations section is appropriate but could be expanded to include:

• potential underreporting or uncertainty in surveillance data incorporated into GBD estimates,

• limitations of ARIMA forecasting for long-term projections,

• the lack of subnational analyses, which may mask regional heterogeneity within China.

Conclusion

The conclusion summarizes the findings but could be made more concise. Some statements regarding policy implications appear stronger than what can be directly supported by the ecological data.

The authors should focus the conclusion on the main epidemiological findings and avoid overly prescriptive policy statements.

The manuscript addresses an important public health topic and uses a widely recognized dataset. However, revisions are needed to improve methodological transparency, interpretation of results, and overall clarity of the manuscript.

Reviewer #2: The author made a thorough analysis of the data and all are also available and easily accessible in the manuscript. The data in the GBD reference, while you may need to log in or create an account to access it, the site is for free and if we need to refer back to these, it is also accessible for anyone. Statistical analysis is also easy to understand through the data presented and is supported through the discussion part. I also appreciate the dedicated definition of terms used by the author in the manuscript. This makes it easier for the reader to have a reference of all the abbreviations used in the manuscript.

6. PLOS authors have the option to publish the peer review history of their article (what does this mean?). If published, this will include your full peer review and any attached files.

Reviewer #1: **Yes:** Nathalia Lopez Duarte

Reviewer #2: **Yes:** Denis Cruz

---

## [Author Response · Author response to Decision Letter 1]

22 Mar 2026

This is a revised submission addressing all reviewer comments. Key changes include: (1) Split Table 1 into two tables; (2) Updated Figure 3 Y-axis labels; (3) Updated GBD 2023 citation; (4) Expanded Limitations section; (5) Refined language throughout. Detailed point-by-point responses to Reviewer #1 and Reviewer #2 are provided in the attached "Response to Reviewers.docx" file.

---

## [Decision Letter · Decision Letter 1]

12 May 2026

HIV/AIDS burden, attributable risk factors, and projections among reproductive-age adults in China, 1990–2035: a GBD 2023 analysis

PONE-D-26-08383R1

Dear Dr. Yan,

We’re pleased to inform you that your manuscript has been judged scientifically suitable for publication and will be formally accepted for publication once it meets all outstanding technical requirements.

Kind regards,

PUGAZHENTHAN THANGARAJU, M.D.,Ph.D., FRCP (LONDON)., FRCP (GLASGOW).,MBA.,

Academic Editor

PLOS One

Additional Editor Comments (optional):

Reviewers' comments:

Reviewer's Responses to Questions

**Comments to the Author**

1. If the authors have adequately addressed your comments raised in a previous round of review and you feel that this manuscript is now acceptable for publication, you may indicate that here to bypass the “Comments to the Author” section, enter your conflict of interest statement in the “Confidential to Editor” section, and submit your "Accept" recommendation.

Reviewer #1: All comments have been addressed

Reviewer #2: All comments have been addressed

2. Is the manuscript technically sound, and do the data support the conclusions?

Reviewer #1: Yes

Reviewer #2: Yes

3. Has the statistical analysis been performed appropriately and rigorously? 

Reviewer #1: Yes

Reviewer #2: Yes

4. Have the authors made all data underlying the findings in their manuscript fully available?

Reviewer #1: Yes

Reviewer #2: Yes

5. Is the manuscript presented in an intelligible fashion and written in standard English?

Reviewer #1: Yes

Reviewer #2: Yes

6. Review Comments to the Author

Reviewer #1: All issues addressed . --------------------------------------------------------------------------------------------------

Reviewer #2: The edits made on the manuscript have made the research more easily digestible. The tables and figures provided supplements well the data discussed in the results section. Extensive data in the figures were also included.

7. PLOS authors have the option to publish the peer review history of their article (what does this mean?). If published, this will include your full peer review and any attached files.

Reviewer #1: **Yes:** Nathalia Lopez Duarte

Reviewer #2: No

---

## [Editor Report · Acceptance letter]

PONE-D-26-08383R1

PLOS One

Dear Dr. Yan,

I'm pleased to inform you that your manuscript has been deemed suitable for publication in PLOS One. Congratulations! Your manuscript is now being handed over to our production team.

Kind regards,

on behalf of

DR. PUGAZHENTHAN THANGARAJU

Academic Editor

PLOS One